# Diffusion Adversarial Post-Training for One-Step Video Generation

**Shanchuan Lin** [1]  **Xin Xia** [1]  **Yuxi Ren** [1]  **Ceyuang Yang** [1]  **Xuefeng Xiao** [1]  **Lu Jiang** [1]

## Abstract

The diffusion models are widely used for image and video generation, but their iterative generation process is slow and expansive. While existing distillation approaches have demonstrated the potential for one-step generation in the image domain, they still suffer from significant quality degradation. In this work, we propose Adversarial Post-Training (APT) against real data following diffusion pre-training for one-step video generation. To improve the training stability and quality, we introduce several improvements to the model architecture and training procedures, along with an approximated R1 regularization objective. Empirically, our experiments show that our adversarial post-trained model can generate two-second, 1280×720, 24fps videos in real-time using a single forward evaluation step. Additionally, our model is capable of generating 1024px images in a single step, achieving quality comparable to state-of-the-art methods. Our project page: https://seaweed-apt.com/

## 1. Introduction

The diffusion model (Ho et al., 2020; Song et al., 2020) has become the de facto method for large-scale image and video generation. Reducing the generation cost is an important research area. Among the various methods, diffusion step distillation has emerged as an effective approach to reduce the inference cost. Generally, these methods start with a pre-trained diffusion model as a teacher that generates targets through multiple diffusion inference steps. They then apply knowledge distillation (Hinton, 2015) to train a student model that can replicate the teacher's output using much fewer diffusion inference steps.

One-step generation is often considered the pinnacle of diffusion step distillation, yet it presents the most significant challenges. It deviates from the fundamental principle of diffusion models, which rely on iterative denoising steps to uncover the data distribution. While previous research has demonstrated notable advancements for generating images in a single step with promising results (Ren et al., 2024; Yin et al., 2024a; Sauer et al., 2025; Lin et al., 2024), producing high-quality images in one step remains challenging, particularly in achieving fine-grained details, minimizing artifacts, and preserving the structural integrity.

Accelerated video generation, however, has seen limited progress in the literature. Early efforts utilizing generative adversarial networks (GANs) (Goodfellow et al., 2014), *e.g.,* StyleGAN-V (Skorokhodov et al., 2021) can only generate domain-specific data with poor quality in modern standards. With the rise of diffusion methods, recent studies have begun exploring the extension of image distillation techniques to video diffusion models. However, existing works have only explored distillation on small-scale and low-resolution video models that only generate 512×512 videos for a total of 16 frames (Lin & Yang, 2024; Zhai et al., 2024). A concurrent work (Yin et al., 2024b) has attempted distillation of large-scale video models at 640×352 12fps. These methods still generally need 4 diffusion steps. Given the prohibitive computational cost associated with high-resolution video generation, *e.g.,* generating just a few seconds of 1280×720 24fps videos can take multiple minutes even on the state-of-the-art GPUs like the H100, our work aims at generating high-resolution videos in a single step.

In this paper, we introduce a new approach for one-step image and video generation. Our method utilizes a pre-trained diffusion model, specifically the diffusion transformer (DiT) (Peebles & Xie, 2023), as initialization, and continues training the DiT using the adversarial training objective against real data. It is important to notice the contrast to existing diffusion distillation methods, which use a pre-trained diffusion model as a distillation teacher to generate the target. Instead, our method performs adversarial training of the DiT directly on real data, using the pre-trained diffusion model only for initialization. We term this method *Adversarial Post-Training* or *APT*, as it parallels supervised fine-tuning commonly performed during the post-training stage. Empirically, we observe that APT provides two benefits. First, APT eliminates the substantial cost associated with pre-computing video samples

---

[1]ByteDance Seed. Correspondence to: Shanchuan Lin <peter-lin@bytedance.com>.

from the diffusion teacher. Second, unlike diffusion distillation, where the quality is inherently constrained by the diffusion teacher, APT demonstrates the ability to surpass the teacher by a large margin in some evaluation criteria, in particular, improving realism, resolving exposure issues, and enhancing fine details.

Direct adversarial training on diffusion models is highly unstable and prone to collapse, particularly in our case, where both the generator and discriminator are large transformer models containing billions of parameters. To tackle this issue, our method introduces several key designs to stabilize training. It incorporates a generator initialized through deterministic distillation and introduces several enhancements to the discriminator, including transformer-based architectural changes, a discriminator ensemble across timesteps, and an approximated R1 regularization loss to stabilize training.

By virtue of APT, we have trained what may be one of the largest GAN ever reported to date (∼16B), capable of generating both images and videos with a single forward evaluation. Our experiments demonstrate that our model achieves overall performance comparable to state-of-the-art one-step image generation methods. More importantly, to the best of our knowledge, our model is the first to demonstrate high-resolution video generation in a single step (1280×720 24fps), surpassing the previous state-of-the-art, which generates 512×512 or 640×352 up to 12fps videos in four steps. On a H100 GPU, our model can generate a two-second 1280×720 24fps video latent using a single step in two seconds. On 8×H100 GPUs with parallelization, the entire pipeline with text encoder and latent decoder runs in real-time.

## 2. Related Works

**Accelerating Diffusion Models.** Diffusion step distillation is a common and effective approach to accelerate diffusion models. Existing methods address this problem using either deterministic or distributional methods.

Deterministic methods exploit the fact that diffusion models learn a deterministic probability flow with exact noise-to-sample mappings and aim to predict the exact teacher output using fewer steps. Prior works include progressive distillation (Salimans & Ho, 2022), consistency distillation (Song et al., 2023; Song & Dhariwal, 2024; Lu & Song, 2024; Luo et al., 2023a;b), and rectified flow (Liu et al., 2023a;b; Yan et al., 2024). Deterministic methods are easy to train using simple regression loss, but the results of few-step generation are very blurry, due to optimization inaccuracy and reduced Lipschitz constant in the student model (Lin et al., 2024). For large-scale text-to-image generation, deterministic methods generally require more steps, *e.g.,* eight steps, to generate desirable samples (Luo et al., 2023a;b).

On the other hand, distributional methods only aim to approximate the same distribution of the diffusion teacher. Most existing methods are researched for image generation. They have severe artifacts for one-step generation and still require multiple steps to obtain desirable results. Notably, LADD (Sauer et al., 2024) and others (Kang et al., 2024; Luo et al., 2024b; Chen et al., 2024) use pre-generated teacher images as the adversarial target. Lightning (Lin et al., 2024) and others (Ren et al., 2024; Kohler et al., 2024; Wang et al., 2024a) learn the teacher trajectory with the adversarial objective in the loop. DMD (Yin et al., 2023) and others (Luo et al., 2024a) apply score distillation (Wang et al., 2023b) from the teacher model. The above methods set the teacher as the upper bound for quality. DMD2 (Yin et al., 2024a) and others (Sauer et al., 2025; Chadebec et al., 2024) apply both adversarial against real data and score distillation from the teacher model. The closest to our work is UFO-Gen (Xu et al., 2023b) which also only applies adversarial training on real data. However, its discriminator adopts the DiffusionGAN (Wang et al., 2022) approach which takes the corrupted data as input. Our method feeds the discriminator with real, uncorrupted data. Hence, our approach follows the standard adversarial training as in GAN more closely. Additionally, UFO-Gen's image generator and discriminator are convolutional models under 1B parameters, while ours are transformer models with 8B parameters and generate both image and video.

**One-Step Video Generation.** One-step video generation works may trace back to the use of generative adversarial networks (Skorokhodov et al., 2021; Clark et al., 2019; Tian et al., 2021; Yu et al., 2022b). They can generate up to 1024px resolution videos but are trained only on domain-restricted data, *e.g.,* talking head videos, and the quality is poor by modern standards. More recently, some distillation works (Lin & Yang, 2024; Wang et al., 2024b; Zhai et al., 2024) have attempted to distill small-scale and low-resolution video models, *i.e.,* AnimateDiff (Guo et al., 2024) and ModelScope (Wang et al., 2023a). These models only generate low-resolution 256px or 512px videos of 16 frames and have substantial quality degradation for one-step generation. For image-to-video generation (I2V), SF-V (Zhang et al., 2024) and OSV (Mao et al., 2024) explore one-step generation of 1024×576 14 frames videos. For text-to-video (T2V) generation, a concurrent work (Yin et al., 2024b) has demonstrated the generation of 640×352 12fps videos in four steps. To the best of our knowledge, our work is the first to demonstrate one-step T2V generation of 1280×720 24fps videos with a duration of 2 seconds (49 frames).

**Stable Adversarial Training.** R1 regularization (Roth et al., 2017) has been shown effective for GAN convergence (Mescheder et al., 2018). It has been used by many prior GANs to improve performance (Karras et al., 2018; 2019; 2021; Kang et al., 2023; Brock et al., 2019; Huang et al.,

2024a). However, many recent distillation works have either completely not used R1 or only used it for parts of the discriminator network. This is likely due to its higher-order gradient computation is computationally expensive and is not supported by modern deep learning software stacks, *i.e.,* FSDP (Zhao et al., 2023), gradient checkpointing (Chen et al., 2016), and FlashAttention (Dao et al., 2022; Dao, 2023; Shah et al., 2024). Our paper proposes an approximation method to address this issue and we find our approximated R1 loss is critical for preventing training collapse.

## 3. Method

Our objective is to convert a text-to-video diffusion model to a one-step generator. We achieve this by fine-tuning the diffusion model with the generative adversarial objective against real data. We refer to this process as *Adversarial Post-Training* or *APT*, due to its resemblance to supervised fine-tuning in the conventional post-training stage.

### 3.1. Overview

We build our method on a pre-trained text-to-video diffusion model (Seawead et al., 2025) capable of generating both images and videos through $T$ diffusion steps. The training follows adversarial optimization that alternates through a min-max game. The discriminator $D$ classifies real samples from generated ones, maximizing $-\mathcal{L}_D$, while the generator $G$ aims to generate samples that fool the discriminator, minimizing $\mathcal{L}_G$. Formerly, we have:

$$\mathcal{L}_D = \mathop{\mathbb{E}}_{\boldsymbol{x},c\sim\mathcal{T}} \big[ f_D(D(\boldsymbol{x},c)) \big] + \mathop{\mathbb{E}}_{\substack{\boldsymbol{z}\sim\mathcal{N}\\c\sim\mathcal{T}}} \big[ f_G(D(G(\boldsymbol{z},c),c)) \big], \quad (1)$$

$$\mathcal{L}_G = \mathop{\mathbb{E}}_{\substack{\boldsymbol{z}\sim\mathcal{N}\\c\sim\mathcal{T}}} \big[ g_G(D(G(\boldsymbol{z},c),c)) \big], \quad (2)$$

where $\mathcal{N}$ denotes the standard Gaussian distribution, and $\mathcal{T}$ represents the training data comprising a paired latent sample $\boldsymbol{x}$ and text condition $c$. The latent and noise samples are of size $\boldsymbol{x}, \boldsymbol{z} \in \mathbb{R}^{t'\times h'\times w'\times c'}$, where $t'$, $h'$, $w'$, $c'$ represent the dimensions of time, height, width, and channel. The functions $f_D$, $f_G$, and $g_G$ are the output functions. Here, we use the simple non-saturating variant (Goodfellow et al., 2014): $f_D(x) = g_G(x) = \log \sigma(x)$ and $f_G(x) = \log(1 - \sigma(x))$, where $\sigma(x)$ is the sigmoid function.

Figure 1 illustrates the overall architecture. Both the generator and the discriminator backbone use the diffusion model architecture but are initialized with different strategies which will be discussed later in this section. Concretely, our diffusion model uses the MMDiT architecture (Esser et al., 2024) and is trained with the flow-matching objective (Lipman et al., 2023) over a mixture of images and videos at their native resolutions (Dehghani et al., 2024) in the latent space (Rombach et al., 2021). The model comprises

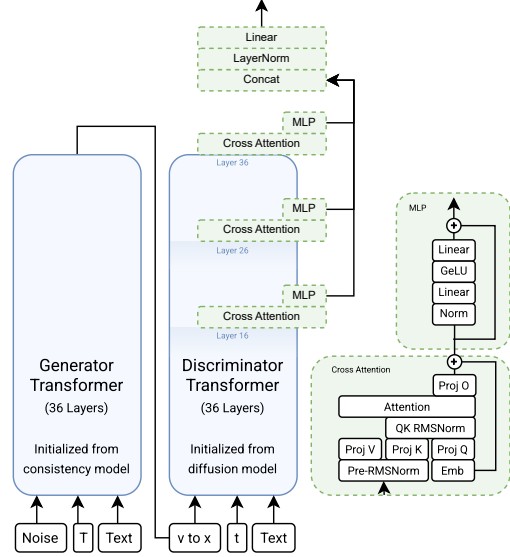

Figure 1: Architecture overview. Both the generator and the discriminator backbone share the diffusion transformer architecture (blue). We add additional output heads on the discriminator network to produce the scalar logit (green).

36 layers of transformer blocks, amounting to a total of 8 billion parameters.

### 3.2. Generator

We find direct adversarial training on the diffusion model leads to collapse. To tackle this, we first employ consistency distillation (Song et al., 2023; Song & Dhariwal, 2024) with mean squared error loss. The model is distilled with a constant classifier-free guidance (Ho & Salimans, 2021) scale of 7.5 and a fixed negative prompt.

Let $\hat{G}$ denote the distilled model. Given noise sample $\boldsymbol{z}$ and text condition $c$, the model $\hat{G}$ predicts the velocity field $\hat{v}$, which can be converted to sample prediction $\hat{\boldsymbol{x}}$:

$$\hat{\boldsymbol{v}} = \hat{G}(\boldsymbol{z},c,T), \quad (3)$$

$$\hat{\boldsymbol{x}} = \boldsymbol{z} - \hat{\boldsymbol{v}}. \quad (4)$$

Although the generated sample $\hat{\boldsymbol{x}}$ is very blurry, $\hat{G}$ provides an effective initialization for the subsequent adversarial training. Therefore, we initialize our generator $G$ with the weights of $\hat{G}$, defined as:

$$G(\boldsymbol{z},c) \coloneqq \boldsymbol{z} - \hat{G}(\boldsymbol{z},c,T). \quad (5)$$

For the subsequent training, we primarily focus on one-step generation capability and always feed the final timestep $T$ to the underlying model.

### 3.3. Discriminator

The discriminator is trained to produce a logit that effectively distinguishes between real samples $\boldsymbol{x}$ and generated

samples $\hat{\boldsymbol{x}}$. In this subsection, we discuss several effective designs that contribute to stable training and quality improvement. Refer to the detailed results presented in our ablation studies.

First, following prior works (Lin et al., 2024; Lin & Yang, 2024; Xu et al., 2023a; Sauer et al., 2024), we initialize the discriminator backbone using the pre-trained diffusion network and let it operate directly in the latent space. Therefore, the discriminator backbone also comprises 36 layers of transformer blocks and 8 billion parameters. We find that training all parameters without freezing improves the quality. Additionally, we find that initializing it with the original diffusion model weights, as opposed to the distilled model weights used by the generator, yields better results.

Second, we modify the diffusion transformer architecture to produce logits. Specifically, we introduce new cross-attention-only transformer blocks at the 16th, 26th, and 36th layers of the transformer backbone. Each block uses a single learnable token as the query to cross-attend to all the visual tokens from the backbone as the key and value, producing a single token output. These tokens are then channel-concatenated, normalized, and projected to yield a single scalar logit output. We find that using features from multiple layers enhances the structure and composition of the generated samples.

Third, we directly provide the discriminator the raw sample $\boldsymbol{x}, \hat{\boldsymbol{x}}$ without any noise corruptions. This avoids the introduction of artifacts to our generated samples. However, since our discriminator backbone is initialized from the diffusion model, and the diffusion pre-training objective at $t = 0$ is not meaningful, we find using $t = 0$ for our discriminator leads to collapse. Therefore, we propose to use an ensemble of different timestep values as input. Specifically, let $\hat{D}$ denote the underlying discriminator model, we define the $D(\boldsymbol{x}, c)$ in Equation (2) as:

$$D(\boldsymbol{x}, c) := \underset{t \sim \text{shift}(\mathcal{U}(0, T), s)}{\mathbb{E}} \left[ \hat{D}(\boldsymbol{x}, t, c) \right], \qquad (6)$$

where $t$ is sampled uniformly from the interval $[0, T]$ and then shifted by transformation function:

$$\text{shift}(t, s) := \frac{s \times t}{1 + (s - 1) \times t}. \qquad (7)$$

The shifting factor $s$ is a hyperparameter determined by the latent dimension $t'$, $h'$, and $w'$. We use $s = 1$ for images and $s = 12$ for videos for our experiment. For efficiency, we sample a single $t$ per training sample $\boldsymbol{x}$ to compute Equation (6).

### 3.4. Regularized Discriminator

Our discriminator, comprising billions of parameters, is prone to collapse. Ensuring stable training is therefore crucial to our problem. The R1 regularization (Roth et al., 2017)

is an effective technique in facilitating the convergence of adversarial training. It penalizes the discriminator gradient $\nabla_{\boldsymbol{x}}$ on real data $\boldsymbol{x}$, preventing the adversarial training from deviating from the Nash-equilibrium:

$$\mathcal{L}_{R1} = \| \nabla_{\boldsymbol{x}} D(\boldsymbol{x}, c) \|_2^2. \qquad (8)$$

Training with R1 requires higher-order gradient computation. The first backward computes the discriminator gradient on input $\nabla_{\boldsymbol{x}}$ as the R1 loss. The second backward computes the gradient of the R1 loss regarding the discriminator parameter for the discriminator updates. However, PyTorch FSDP (Zhao et al., 2023), gradient checkpointing (Chen et al., 2016), FlashAttention (Dao et al., 2022; Dao, 2023; Shah et al., 2024), and other fused operators (Nvidia-Apex) do not support higher-order gradient computation or double backward at the time of writing, preventing the use of R1 in large-scale transformer models.

We propose an approximated R1 loss, written as:

$$\mathcal{L}_{aR1} = \| D(\boldsymbol{x}, c) - D(\mathcal{N}(\boldsymbol{x}, \sigma \mathbf{I}), c) \|_2^2. \qquad (9)$$

Specifically, we perturb the real data with Gaussian noise of small variance $\sigma$. The loss encourages the discriminator's predictions to be close between the real data and its perturbation, thereby reducing the discriminator gradient on real data and achieving a consistent objective as the original R1 regularization. Thus, the final discriminator loss $\mathcal{L}_D$ is:

$$\mathcal{L}_D = \underset{\boldsymbol{x}, c \sim \mathcal{T}}{\mathbb{E}} \left[ f_D(D(\boldsymbol{x}, c)) \right] + \underset{\substack{\boldsymbol{z} \sim \mathcal{N} \\ c \sim \mathcal{T}}}{\mathbb{E}} \left[ f_G(D(G(\boldsymbol{z}, c), c)) \right]$$
$$+ \lambda \underset{\boldsymbol{x}, c \sim \mathcal{T}}{\mathbb{E}} \left[ \| D(\boldsymbol{x}, c) - D(\mathcal{N}(\boldsymbol{x}, \sigma \mathbf{I}), c) \|_2^2 \right]. \qquad (10)$$

In our experiments, we use $\lambda = 100$, $\sigma = 0.01$ for images and $\sigma = 0.1$ for videos. The approximated R1 is applied on every discriminator step.

### 3.5. Training Details

We first train the model on only 1024px images. We use $128 \sim 256$ H100 GPUs with a batch size of 9062. The learning rate is 5e$-$6 for both the generator and the discriminator. We find the model adapts quickly. We use an Exponential Moving Average (EMA) decay rate of 0.995 and adopt the EMA checkpoint after 350 updates on the generator before the quality starts to degrade.

We then train the model on only 1280×720 24fps videos. The videos are clipped to 2 seconds. The generator is initialized from the image EMA checkpoint. The discriminator is re-initialized from the diffusion weights. We use 1024 H100 GPUs with gradient accumulation to reach a batch size of 2048. We lower the learning rate to 3e$-$6 for stability and train it for 300 updates. We find that the one-step model can also perform zero-shot two-step inference with improved details, but more steps lead to artifacts.

| Diffusion | APT | Diffusion | APT |
|---|---|---|---|

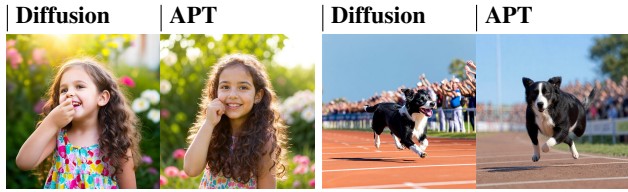

Figure 2: Our 1-step APT model can sometimes generate more realistic tones and better details compared to 25-step diffusion with CFG.

RMSProp optimizer is used with $\alpha = 0.9$, which is equivalent to Adam (Kingma & Ba, 2014) with $\beta_1 = 0, \beta_2 = 0.9$ with reduced memory consumption. We do not use weight decay or gradient clipping. The entire training is conducted in BF16 mixed precision. We use the same datasets as the original diffusion model.

# 4. Experimental Results

This section empirically verifies the proposed *Adversarial Post-Training (APT)* method. Section 4.1 provides a qualitative comparison of our method against other one-step image generation baselines and an analysis of the characteristics of the results generated by our approach. Section 4.2 presents several user studies that quantitatively assess our method. More results are in supplementary materials.

**Baseline.** For comparison with one-step image generation methods, we select FLUX-Schnell (FLUX-Schnell), SD3.5-Turbo (Sauer et al., 2024), SDXL-DMD2 (Yin et al., 2024a), SDXL-Hyper (Ren et al., 2024), SDXL-Lightning (Lin et al., 2024), SDXL-Nitro (Chen et al., 2024), and SDXL-Turbo (Sauer et al., 2025) as the comparison baselines. These models are selected because they are either the latest research publications or commonly available open-source distilled models. We also compare their original diffusion models, *i.e.,* FLUX (FLUX), SD3 (Esser et al., 2024), and SDXL (Podell et al., 2023), against ours in 25 Euler steps. We use the default classifier-free guidance (CFG) (Ho & Salimans, 2021) setting of each model as configured in diffusers (von Platen et al., 2022), while ours uses CFG 7.5. All models are 1024px, except SDXL-Turbo is 512px.

## 4.1. Qualitative Evaluation

For image generation, we first compare our APT's one step with our original diffusion model's 25 steps in Figure 2. We observe that the diffusion model with CFG often generates over-exposed images, rendering the images appear synthetic. In comparison, the APT model tends to generate images with a more realistic tone. Figure 3 further compares our method with other one-step image generation methods. Our method shows advantages in preserving details and

structural integrity.

For video generation, Figure 4 compares our APT one-step and two-step results with the original diffusion's 25-step results. Both the good and the bad cases of the APT method are displayed. For good cases, APT improves details and realism. The one- or two-step APT models still perform worse in terms of structural integrity and text alignment compared to 25-step diffusion. Videos are in supplementary materials.

## 4.2. User Study

**Evaluation Protocol.** We conduct a series of user studies with respect to three criteria: *visual fidelity*, *structural integrity*, and *text alignment*. Specifically, visual fidelity accounts for texture, details, color, exposure, and realism; structural integrity focuses on the structural correctness of the objects and body parts; text alignment measures closeness to the conditional prompts. Human raters are shown pairs of samples generated by different models and asked to choose their preferences regarding each criterion or to indicate no preference if a decision cannot be made.

Afterward, the preference score is calculated as $(G - B)/(G + S + B)$, where $G$ denotes the number of good samples preferred, $B$ denotes the number of bad samples not preferred, and $S$ denotes the number of similar samples without preference. Thus, a score of $0\%$ represents equal preference between the two models. $+100\%$ represents the model is preferred over all evaluated samples, and vice versa for $-100\%$.

For image evaluation, we follow the evaluation protocol in previous diffusion distillation works (Sauer et al., 2024; 2025) and generate samples using 300 randomly selected prompts from PartiPrompt (Yu et al., 2022a) and DrawBench (Saharia et al., 2022). We generate 3 images per prompt. For video evaluation, we generate one video per 96 custom prompts. We have 3 raters to evaluate each comparison pair. The entire user study takes 50,328 sample comparisons.

Additionally, following the previous works (Lin et al., 2024; Sauer et al., 2024; 2025), we also report the FID (Heusel et al., 2017), PFID (Lin et al., 2024), and CLIP (Radford et al., 2021) metrics on COCO dataset (Lin et al., 2014). Note that we find these automatic metrics to be less accurate than user studies for assessing the model's actual performance. We provide the results and discussion in Appendix A. We also provide video quantitative metrics on VBench (Huang et al., 2024b) in Appendix B.

**Image Generation: One-Step *v.s.* 25-Step.** We first compare all one-step model against their corresponding original diffusion model in 25 steps in Table 1. The table shows that all existing one-step methods have degradation in all

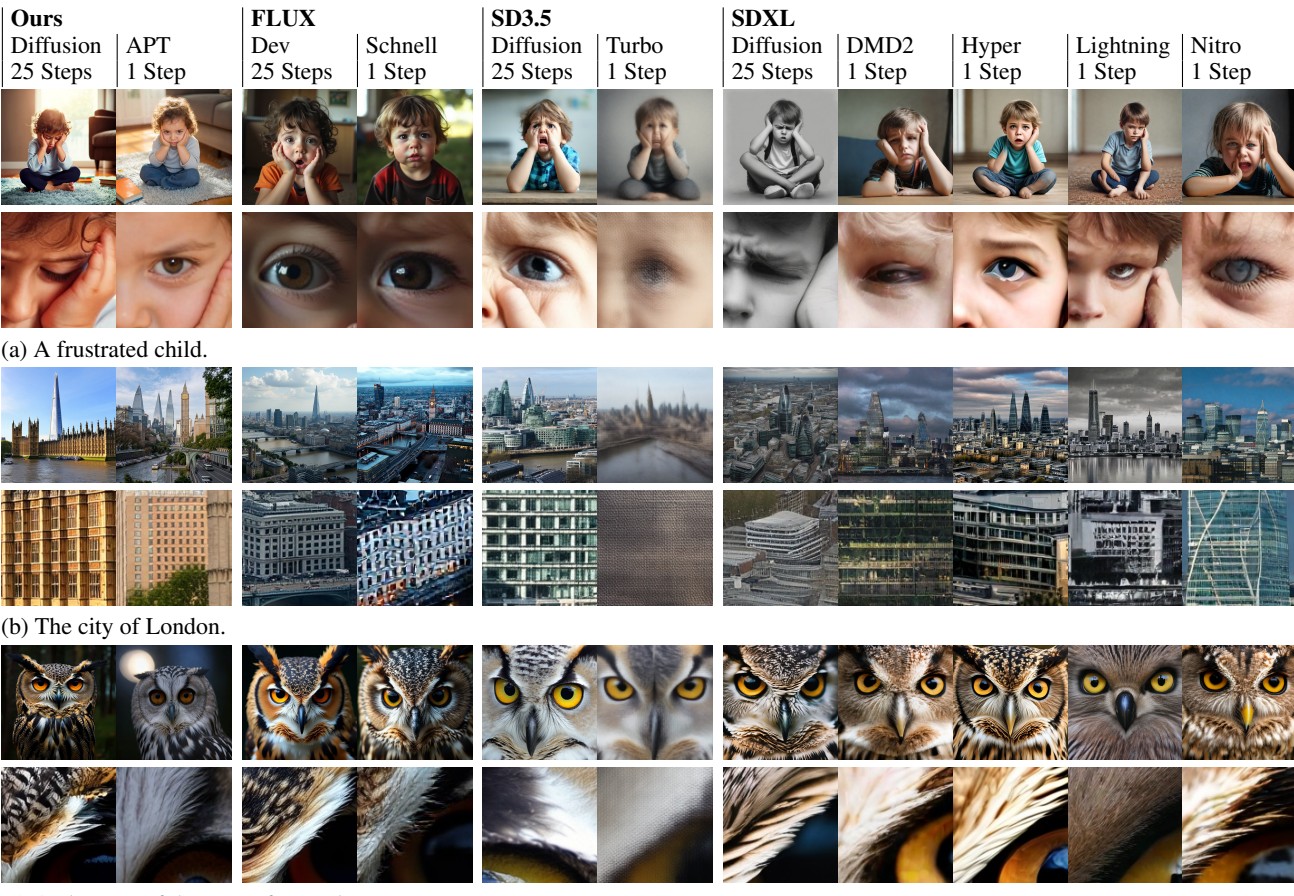

(a) A frustrated child.

(b) The city of London.

(c) A close-up of the eyes of an owl.

Figure 3: Image generation comparison across methods and models. We show results of 1-step generation and the corresponding diffusion model 25-step generation. Our method is significantly better in image details and is among the best in structural integrity. More results are in **??**.

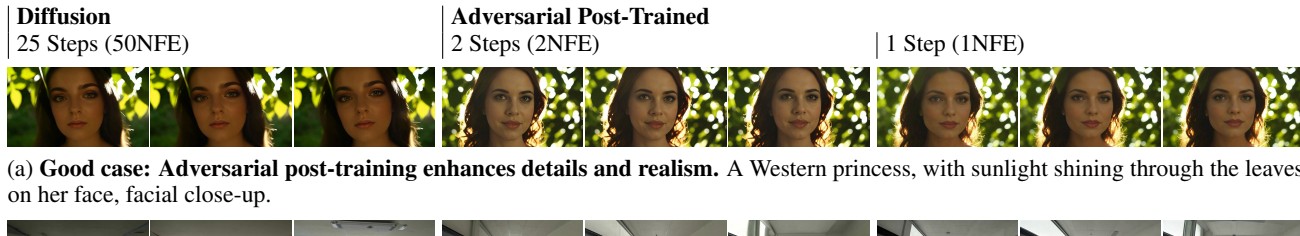

(a) **Good case: Adversarial post-training enhances details and realism.** A Western princess, with sunlight shining through the leaves on her face, facial close-up.

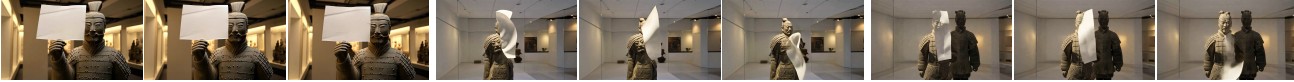

(b) **Average case: Adversarial post-training can produce the scene but with degradation in structure and text alignment.** First-person perspective, the camera passes through a classroom entering the school playground.

(c) **Failure case: Adversarial post-training can fail at some prompts.** A terracotta warrior holds a white paper in one hand, and the paper flutters in the wind. The background is a museum.

Figure 4: Video generation results. Adversarial post-training can improve visual fidelity, *i.e.,* details and realism, but few-step generation still has degradation in structure and text alignment. More full videos are provided in **??**.

Table 1: One-step image generation compared to their corresponding original diffusion models in 25 steps.

| Image One-Step *v.s.* Diffusion 1 Step *v.s.* 25 Steps | Visual Fidelity | Structural Integrity | Text Align |
|---|---|---|---|
| FLUX-Schnell | -36.6% | -24.4% | -2.8% |
| SD3.5-Large-Turbo | -94.4% | -30.1% | -20.4% |
| SDXL-DMD2 | -9.3% | -16.8% | -4.6% |
| SDXL-Hyper | -8.8% | -12.3% | -2.1% |
| SDXL-Lightning | -7.7% | -15.1% | -17.4% |
| SDXL-Nitro-Realism | -21.6% | -22.7% | -5.6% |
| SDXL-Turbo | -80.1% | -14.9% | -1.2% |
| APT (Ours) | +37.2% | -13.1% | -8.1% |

Table 2: Comparison to the state-of-the-art one-step image generation. Both absolute and relative preference scores (adjusted for base model performance) are presented. The methods are sorted by average preference.

| Image Ours *v.s.* Others 1 Step | Visual Fidelity | Structural Integrity | Text Align | Average |
|---|---|---|---|---|
| **Absolute** | | | | |
| FLUX-Schnell | +35.7% | -21.5% | -28.1% | -4.6% |
| SDXL-DMD2 | +34.7% | +10.3% | -11.8% | +11.1% |
| SDXL-Nitro-Realism | +24.6% | +16.7% | -4.9% | +12.1% |
| SDXL-Hyper | +43.6% | +4.1% | -6.7% | +13.7% |
| SDXL-Lightning | +34.1% | +14.1% | +11.4% | +19.9% |
| SDXL-Turbo | +68.9% | +14.9% | -7.9% | +25.3% |
| SD3.5-Large-Turbo | +97.8% | +7.7% | -16.7% | +29.6% |
| **Relative** | | | | |
| SDXL-DMD2 | +22.6% | +9.1% | -12.0% | +6.6% |
| SDXL-Nitro-Realism | +12.5% | +15.5% | -5.1% | +7.6% |
| SDXL-Hyper | +31.5% | +2.9% | -6.9% | +9.2% |
| SDXL-Lightning | +22.0% | +12.9% | +11.2% | +15.4% |
| SDXL-Turbo | +56.8% | +13.7% | -8.1% | +20.8% |
| FLUX-Schnell | +77.1% | +11.4% | -9.2% | +26.4% |
| SD3.5-Large-Turbo | +134.0% | +33.3% | -2.5% | +54.9% |

three criteria. In terms of structural integrity, our method has degradation but is less than almost all existing methods except for SDXL-Hyper. Our method is weaker in text alignment but is still mid-tier. It is worth noting that our model is the only one to achieve a more favorable evaluation criterion (visual fidelity), aligning with our qualitative observation that APT enhances details and realism. The improvement over the original diffusion model can be attributed to our method's approach, which forgoes using the diffusion model as a teacher and instead performs direct adversarial training on real data. More discussions are provided in Section 5.3.

**One-Step Image Generation: Comparison to the State-of-the-Art.** In Table 2, we compare our one-step generation to the state-of-the-art one-step image generation models. We show both the absolute preference score and the adjusted relative change based on their corresponding diffusion model baseline, which will be described later.

The results in Table 2 demonstrate that our method achieves

Table 3: Comparison of other base diffusion models with our base model for image generation. All models are evaluated using 25 steps. Since our model is designed to handle both video and image generation, its image generation performance is slightly inferior to FLUX and SD3.5.

| Image Ours *v.s.* Others 25 Steps | Visual Fidelity | Structural Integrity | Text Align |
|---|---|---|---|
| FLUX | -41.4% | -32.9% | -18.9% |
| SD3.5-Large | -36.2% | -25.6% | -14.2% |
| SDXL | +12.1% | +1.2% | +0.2% |

Table 4: Comparison of one-step (and two-step) video generation with the original 25-step diffusion models.

| Video 1 and 2 Steps *v.s.* 25 Steps | Steps | Visual Fidelity | Structural Integrity | Text Align |
|---|---|---|---|---|
| APT (Ours) | 2 | +32.3% | -31.3% | -9.4% |
| | 1 | +10.4% | -38.5% | -8.3% |

performance comparable to the state-of-the-art in one-step image generation. On average, it ranks second in absolute preference, trailing FLUX-Schnell, and ranks first in relative preference. Compared to the baseline methods, our model is preferred for its visual fidelity and structural integrity but is less preferred in text alignment. The weaker text alignment is a limitation of the APT method. Further discussion on the causes can be found in Section 5.5.

The relative preference score is introduced to reduce the bias that a strong base model can unfairly influence the evaluation of the one-step acceleration method. Thus we report the base model rates in Table 3. Because our base model handles both video and image generation, uses different training data, and has not undergone human-preference tuning, our diffusion model is weaker compared to FLUX and SD3.5. We highlight these differences in the base model for interpreting the comparison of one-step image generation in Table 2.

**One-step Video Generation.** Table 4 compares the one-step and two-step video generation results compared to the original diffusion baseline using 25 steps. The trend is similar to the image performance, where our model outperforms the original diffusion model in visual fidelity, but has degradation in structural integrity and text alignment. Despite the degradation, the videos generated in one step maintain decent quality at 1280×720 resolution. We refer readers to view the videos on our website. The degradation in structural integrity appears to be more severe in videos compared to images since it now involves motions. Our work is a preliminary proof of concept. We emphasize the need for further research to advance one-step video generation.

# 5. Ablation Study and Discussion

## 5.1. The Effect of Approximated R1 Regularization

We find that the approximated R1 regularization is critical for maintaining stable training. Without this regularization, training collapses rapidly. As shown in Figure 5, the black curve represents the discriminator loss without regularization, which quickly approaches zero compared to the green curve that includes the regularization. When the discriminator loss approaches zero, the generator produces colored plates, as depicted on the right side of Figure 5.

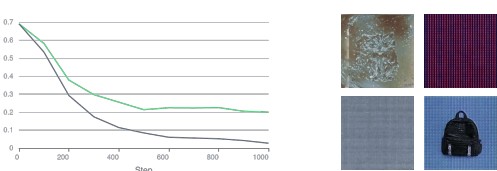

Figure 5: Without approximated R1 regularization, the discriminator loss reaches zero (black) and the training collapses. With approximated R1 regularization, the discriminator loss does not reach zero (green).

## 5.2. Discriminator Design

We first experiment with using different depths of the pretrained diffusion model as our discriminator. Figure 6 shows that a deeper discriminator leads to better image quality.

| Half-Depth Discriminator Layer 14, 16, 18th | Two-Third-Depth Discriminator Layer 18, 22, 26th | Full-Depth Discriminator Layer 16, 26, 36th |
| --- | --- | --- |

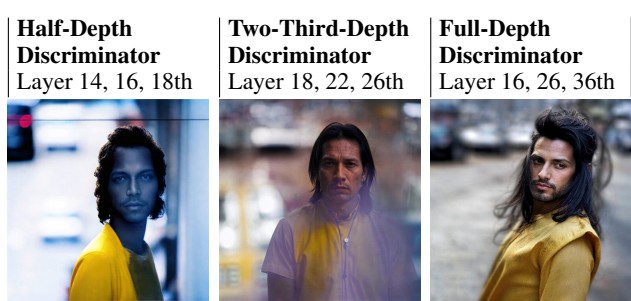

Figure 6: A deeper discriminator that includes the full depth of the pre-trained network leads to better quality.

We then verify the effectiveness of using multilayer features. Figure 7 shows that using multilayer features can improve structural correctness.

| Last-Layer Discriminator Layer 36th | Multi-Layer Discriminator Layer 16, 26, 36th |
| --- | --- |

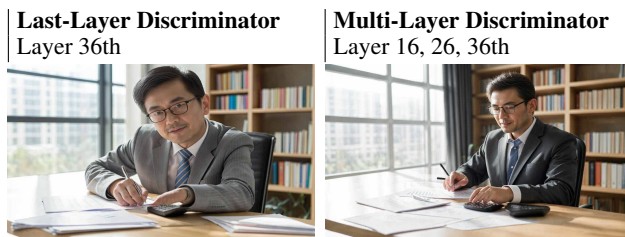

Figure 7: The multi-layer discriminator improves structural integrity.

## 5.3. The Reason for Visual Improvement

We find our APT model mainly resolves the over-exposure and over-saturating tone issue resulting from CFG. We find APT can help the generator learn distributions closer to the real data. More discussions are in Appendix H.

## 5.4. The Cause of Structural Degradation

We take the one-step image model and interpolate the input noise $z$ to generate latent traversal visualizations. We find structural incorrectness often occurs during the smooth transition between modes, where the diffusion model switches between modes more sharply. We hypothesize the generator being under-capacitated is one of the main causes of the degradation in structural integrity. More discussions are in Appendix I.

| Mode A | Transitioning | Mode B |
| --- | --- | --- |

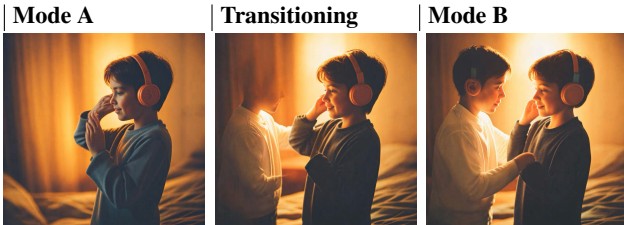

Figure 8: One-step generator has limited capacity to make sharp switches between modes.

## 5.5. Analysis of Text Alignment Degradation

Previous research has found CFG can force the generated distribution to have even stronger text alignment than the real data distribution, *i.e.,* samples looking canonical (Karras et al., 2024; Lin & Yang, 2023). APT learns from the real distribution without CFG's boosting effect. Our dataset has been re-captioned, so our APT model still has acceptable text alignment, though not as strong as CFG. Furthermore, the problem in Section 5.4 also has a negative impact on the alignment of the text. We have conducted additional experiments and have provided more discussions in Appendix J.

# 6. Conclusion and Limitations

We propose Adversarial Post-Training (APT) for single-step generation of both image and video. Our method incorporates several architectural enhancements and an approximate R1 regularization, which are crucial for training stability.

To the best of our knowledge, we present the first proof of concept for generating high-resolution videos (1280×720 24fps) in a single step. However, we identify several limitations in the current approach. First, due to computational constraints, we were only able to train the model to generate videos for up to two seconds. Second, the one-step generation still has degradation in structural integrity and text alignment, which we aim to address in future works.

## Acknowledgment

We thank Jiashi Li, Zuquan Song, and Junru Zheng for managing the compute infrastructure. We thank Yanzuo Lu, Meng Guo, Weiying Sun, Shanshan Liu and Yameng Li for image evaluation. We thank Ruiqi Xia, Donglei Ji, Juntian Chen, and Songyan Yao for video evaluation.

## Impact Statement

This paper presents work whose goal is to advance the field of Machine Learning. There are many potential societal consequences of our work, none which we feel must be specifically highlighted here.

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

## A. Image Quantitative Metrics

Following the previous works (Sauer et al., 2025; Chen et al., 2024; Yin et al., 2024a; Lin et al., 2024), we calculate Fréchet Inception Distance (FID) (Heusel et al., 2017), Patch Fréchet Inception Distance (PFID) (Lin et al., 2024), and CLIP score (Radford et al., 2021) on COCO dataset (Lin et al., 2014). We notice some of the works use COCO-5K (Sauer et al., 2025; Chen et al., 2024) while others use COCO-10K (Lin et al., 2024; Yin et al., 2024a). We provide both in Tables 5 and 6. Specifically, we take the first 5,000 and 10,000 captions from the COCO 2014 validation dataset as the generation prompts. FID is calculated between the generated images and the ground-truth COCO images. The ground-truth images are cropped to a square aspect ratio before resizing to 299px for the inception network (Szegedy et al., 2015). Resizing is done properly with established library (Parmar et al., 2021). Patch FID (PFID) metrics (Lin et al., 2024; Yin et al., 2024a) center-crops 299px patches without resizing to measure image details. For 512px models, the results are bilinear upsampled to 1024px before center-crop following prior work (Lin et al., 2024). For the CLIP score, we follow the previous works (Sauer et al., 2025) to use the laion/CLIP-ViT-g-14-laion2B-s12B-b42K model. We use each model's default CFG (Ho & Salimans, 2021) as configured in diffusers (von Platen et al., 2022), *e.g.,* FLUX uses 3.5 baked-in CFG. Our diffusion model uses 7.5 CFG. We also verify that our metric calculations match what was reported in the original publications, *e.g.,* DMD2's reported FID 19.01 is close to our report of 18.10. SDXL-Lightning's reported FID 22.61 is close to our report of 23.40, *etc.* The difference can be attributed to the randomness of the seed and the selection of prompts.

We find that the metrics can be far off from the model's actual performance. For example, FLUX 1-step surpasses FLUX 25-step in all metrics. However, this strongly contradicts our human perception and evaluation in Table 1, and our observation that FLUX 1-step has visible degradation in visual quality and structural integrity. Furthermore, the metrics suggest that SDXL 25-step is better than FLUX 25-step in all metrics. This also contradicts our human evaluation in Table 3 and our observation that SDXL should have a weaker performance than FLUX.

We notice that most previous diffusion distillation publications (Sauer et al., 2024; 2025; Lin et al., 2024; Ren et al., 2024; Chen et al., 2024) are conducted on the SDXL model. The metrics within the SDXL family of models appear more reasonable, potentially concealing the issue. Yet, this does not mean that the metrics are valid within the same architecture family of models, as the metrics wrongly identify FLUX 1-step as being better than FLUX 25-step.

Therefore, we primarily rely on human evaluations for our work. We caution researchers to interpret these metrics

Table 5: Quantitative metrics on COCO5K across different models and methods for image generation. We find the metrics inaccurately capture the actual performance of the model. Discussion are provided in Appendix A.

| Method | Steps | FID↓ | PFID↓ | CLIP↑ |
|---|---|---|---|---|
| FLUX-Dev | 25 | 31.8 | 38.7 | 32.9 |
| FLUX-Schnell | 1 | 24.9 | 30.0 | 33.7 |
| SD3.5-Large | 25 | 25.1 | 30.8 | 33.8 |
| SD3.5-Turbo | 1 | 61.6 | 174.5 | 30.4 |
| SDXL | 25 | 25.1 | 28.7 | 33.9 |
| SDXL-DMD2 | 1 | 24.1 | 33.0 | 34.1 |
| SDXL-Hyper | 1 | 36.9 | 41.8 | 33.1 |
| SDXL-Lightning | 1 | 28.9 | 36.9 | 31.9 |
| SDXL-Nitro-Realism | 1 | 26.4 | 32.8 | 33.9 |
| SDXL-Turbo (512px) | 1 | 28.4 | 66.0 | 33.8 |
| Our Diffusion | 25 | 26.9 | 30.6 | 33.2 |
| Our Consistency | 1 | 119.6 | 164.6 | 22.3 |
| Our APT | 1 | 27.9 | 34.6 | 32.3 |

Table 6: Quantitative metrics on COCO10K for image generation. The metrics also inaccurately capture the actual performance of the model. Discussion are provided in Appendix A.

| Method | Steps | FID↓ | PFID↓ | CLIP↑ |
|---|---|---|---|---|
| FLUX-Dev | 25 | 26.3 | 32.8 | 32.9 |
| FLUX-Schnell | 1 | 18.8 | 23.7 | 33.7 |
| SD3.5-Large | 25 | 19.2 | 23.7 | 33.8 |
| SD3.5-Turbo | 1 | 55.7 | 170.9 | 30.4 |
| SDXL | 25 | 19.2 | 22.5 | 33.9 |
| SDXL-DMD2 | 1 | 18.1 | 26.2 | 34.0 |
| SDXL-Hyper | 1 | 31.4 | 35.3 | 33.2 |
| SDXL-Lightning | 1 | 23.4 | 30.4 | 31.9 |
| SDXL-Nitro-Realism | 1 | 20.2 | 26.2 | 33.9 |
| SDXL-Turbo (512px) | 1 | 22.8 | 35.7 | 33.8 |
| Our Diffusion | 25 | 20.7 | 24.7 | 33.1 |
| Our Consistency | 1 | 114.1 | 161.3 | 22.3 |
| Our APT | 1 | 22.1 | 28.5 | 32.3 |

carefully, and we leave the exploration for better metrics to future works.

## B. Video Quantitative Metrics

For video generation, we also provide VBench (Huang et al., 2024b) quantitative metrics in VBench in Table 7. Specifically, we generate 5 videos for each of VBench's 946 prompts and report the VBench scores. As the table shows, APT significantly outperforms the baseline consistency distillation 1NFE, 2NFE, and 4NFE. We observe that the consistency 1NFE baseline generates lower-quality results (e.g., blurry videos), and these issues persist even when increasing to 4NFE. These metrics verify the effectiveness of APT post-training compared to the baseline. Even when

compared to our base model at 50NFE (25 steps + CFG), despite this being an unfair comparison, our APT 1NFE achieves a comparable total score (82.00 vs. 82.15).

Table 7: Quantitative metrics on VBench for video generation. Comparison between our APT model and the diffusion/consistency baselines.

| Method | Steps | Total Score | Quality Score | Semantic Score |
|---|---|---|---|---|
| Diffusion | 25 | 82.15 | 84.36 | 73.31 |
| Consistency | 4 | 77.97 | 81.93 | 62.10 |
| | 2 | 74.20 | 78.83 | 55.69 |
| | 1 | 67.05 | 73.78 | 40.15 |
| APT | 2 | 81.85 | 84.39 | 71.70 |
| | 1 | 82.00 | 84.21 | 73.15 |

## C. Inference Speed

Table 8 shows the inference speed of our model on different numbers of H100 GPUs with parallelization. Our model can generate a 1280×720 24fps 2-second video in 6.03s on a single H100 GPU. With 8 H100 GPUs, it can achieve real-time. More optimization can be applied to make it more efficient.

Table 8: Inference speed under different numbers of H100 GPUs for one-step two-second 1280×720 24fps video generation.

| # of H100 | Component | Seconds |
|---|---|---|
| 1 | Text Encoder | 0.28 |
| | DiT | 2.65 |
| | VAE | 3.10 |
| | **Total** | **6.03** |
| 4 | Text Encoder (no parallelization) | 0.28 |
| | DiT | 0.73 |
| | VAE | 1.19 |
| | **Total** | **2.20** |
| 8 | Text Encoder (no parallelization) | 0.28 |
| | DiT | 0.50 |
| | VAE (only 4-GPU parallelization) | 1.19 |
| | **Total** | **1.97** |

## D. The Effect of Training Iterations and EMA

Figure 9 shows the model adapts fast. For the non-EMA model, even after 50 updates, it is able to generate sharp images. The EMA model generally performs better than the non-EMA. We find the quality peaks at 350 updates for the EMA model, and training it longer leads to more structural degradation.

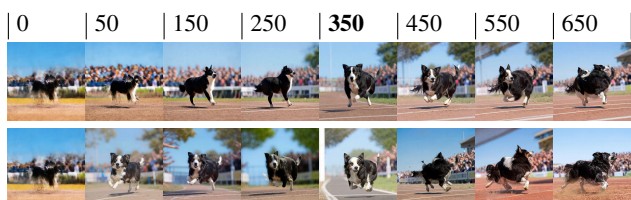

Figure 9: Training progression measured by generator updates. (EMA top, non-EMA bottom)

## E. The Effect of the Batch Size

For images, our early experiments suggest that a larger batch size improves stability and structural integrity, confirming with previous research (Xu et al., 2023b; Kang et al., 2023). For videos, we find that using a small batch size of 256 leads to mode collapse as shown in Figure 10 whereas a large batch size of 1024 does not. Therefore, our final training adopts a large batch size of 9062 for images and a batch size of 2048 for videos.

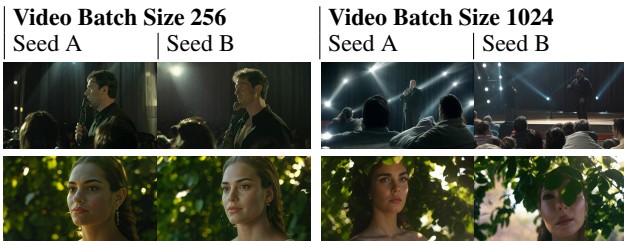

Figure 10: A large batch size prevents mode collapse. A small batch size has mode-collapsed across prompts and seeds.

## F. The Effect of the Learning Rate

We first search for learning rates in image settings. We find that 1e-6 and 1e-5 are either too small or too large and settle on 5e-6 for stable training. Then we experiment with freezing or lowering the learning rate for the discriminator backbone but find that it is best to train the entire discriminator network, as shown in Figure 11.

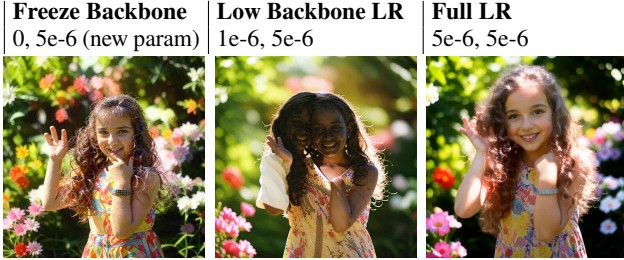

Figure 11: It is better to train the entire discriminator than to freeze or lower the learning rate of the backbone. Freezing the backbone produces undesirable artifacts (zoom in for detail). Lowering the backbone learning rate causes slow convergence.

## G. Understanding the Model Internals

We freeze the model and add an additional linear projection on every layer. It is trained to match the final layer's latent prediction with mean squared error loss. This helps us visualize the internals of our model. As Figure 12 shows, the network's shallow layers generate the coarse structure and the deeper layers generate the high-frequency details. This is similar to the iterative generation process of diffusion models, except in our model the entire generation process is compressed within the 36 transformer layers in a single forward pass.

| Layer 6 | Layer 12 | Layer 18 | Layer 24 | Layer 30 | Layer 36 |

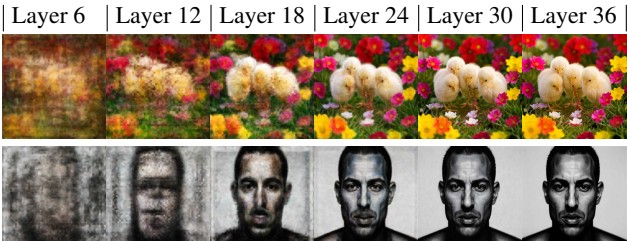

Figure 12: The intermediate result of each transformer layer inside the one-step image generation model.

## H. More Discussion on Visual Improvement

Diffusion models without classifier-free guidance (CFG) (Ho & Salimans, 2021) generate very poor samples (Lin & Yang, 2023). CFG is used ubiquitously to boost perceptual quality and text alignment, but recent works have shown that it can push the generated distribution away from the training distribution, resulting in samples that appear synthetic, over-saturated, and canonical (Lin et al., 2023; Karras et al., 2024). A recent work (Lin & Yang, 2023) has elucidated that the cause of diffusion models generating poor samples without CFG may be rooted in the mean squared error (MSE) loss objective. It has also demonstrated the use of perceptual loss can better learn the real data distribution. We hypothesize that adversarial training can be viewed as an extension of this work, where it does not have the MSE issue and the discriminator is a learnable, in-the-loop, perceptual critic. This helps the generator to learn distributions closer to the real training data.

## I. More Discussion on Structural Degradation

We take the one-step image model and interpolate the input noise $z$ to generate latent traversal videos. The videos are available on the webpage described in **??**. Unlike GAN models which normally have a low-dimensional noise $z$, our model has a very high-dimensional $z \in \mathbb{R}^{t' \times h' \times w' \times c'}$. We find interpolation on the high-dimensional $z$ still produces traversal videos with semantic morphing. Compared to the diffusion model which switches between modes very quickly, our one-step generation model has a much smoother transition between modes. This is likely because the one-step model effectively is much shallower in depth, has less nonlinearity, and has a lower capacity for making drastic changes. This effect has also been observed by a prior work (Lin et al., 2024). We hypothesize the generator being under-capacitated is one of the main causes of the degradation in structural integrity. As Figure 8 shows, we find that structural incorrectness often occurs during the transition between modes. This negatively affects the text alignment, as hallucinations may occur, *e.g.,* one object might appear as two during the transition phase. The training loss supports this hypothesis, where the discriminator can always differentiate before convergence. We aim to address this issue in future works.

## J. Additional Explorations on Text Alignment

We have explored several techniques to improve text alignment but found them to be ineffective. These include: (1) providing the discriminator with unmatched conditional pairs to penalize misalignment, as proposed by (Kang et al., 2023; Sauer et al., 2023). However, we choose not to adopt this approach because our early experiments showed minimal improvements; (2) incorporating CLIP loss (Radford et al., 2021). Our experiments indicate that it could negatively impact visual fidelity, leading to poorer details and artifacts. We leave further investigation to future research.

