# OpenReview forum: "Diffusion Adversarial Post-Training for One-Step Video Generation"
_ICML.cc/2025/Conference — ICML 2025 poster_

### Official Review · Reviewer_afjC · 2025-03-12

**Overall Recommendation:** 2

**Summary:**

This paper proposes an adversarial finetuning framework for one-step T2I and T2V generation for the flow denosing model, which introduces the insights of designing the discriminator and the model and how to stabilize the model training. And it achieves an exciting performance compared with other few-step models.

**Claims And Evidence:**

yes

**Essential References Not Discussed:**

The references are sufficient.

**Experimental Designs Or Analyses:**

The experiments seem to be not efficient; the main focus of this paper is on the one-step video generation. However, there are not many experiments on the analyses of the video part, but it focuses on the image models.

**Methods And Evaluation Criteria:**

yes

**Other Comments Or Suggestions:**

Please add more experimental comparisons with other video models, at least the base model and the one-step mode on the public benchmark.

**Other Strengths And Weaknesses:**

The strengths of this paper can be concluded as 1. concrete design of adversarial diffusion finetuning for most recent diffusion models with DiT architecture and the flow matching objective. 2. The simplified alternative R1 for stabilizing the model training.

The weakness of this paper is that there is not much experimental comparison for the video generation. 720p one step generation is exciting, but 5s of 1 step video generation would be so great.

**Questions For Authors:**

Please see above comments.

**Relation To Broader Scientific Literature:**

The key ideas of this paper are most related to the UFOGEN (https://arxiv.org/abs/2311.09257), which presents the adversarial fine-tuning for diffusion models on 1-step denosing generation.

**Theoretical Claims:**

Yes, they look good to me.

---

> ### Author Rebuttal · Authors · 2025-04-01
>
> **Video Comparisons**
>
> We would like to clarify that existing research on diffusion acceleration has primarily focused on the image domain. To the best of our knowledge, no prior studies have proposed high-resolution, one-step video generation methods, and consequently, no suitable baselines exist for comparison. Therefore, in our submission, we evaluate our method using the image generation task, for which comparable prior works are available.
>
>
> We acknowledge the reviewer’s perspective. Following the reviewer’s suggestion, we have included a comparison with the consistency distillation baseline and our base model.
> The evaluation follows the public [VBench](https://vchitect.github.io/VBench-project/) protocol. Specifically, we generate 5 videos for each of VBench's 946 prompts and report the VBench scores in the table below.
>
>
> As the table shows, APT significantly outperforms the baseline consistency distillation 1NFE, 2NFE, and 4NFE. We observe that the consistency 1NFE baseline generates lower-quality results (e.g., blurry videos), and these issues persist even when increasing to 4NFE. This result verify the effectiveness of APT post-training compared to the baseline.
>
> Even when compared to our base model at 50NFE (25 steps + CFG)—despite this being an unfair comparison—our APT 1NFE achieves a comparable total score (82.00 vs. 82.15) and even performs better on some metrics.
>
>
> We will add the table and the analysis in the revised paper.
>
> | *VBench* | Total Score | Quality Score | Semantic Score | Multiple Objects | Overall Consistency | Spatial Relationship | Temporal Style | Object Class | Dynamic Degree | Aesthetic Quality | Human Action | Scene | Imaging Quality | Background Consistency | Subject Consistency | Temporal Flickering | Motion Smoothness | Color | Appearance Style |
> |---|---|---|---|---|---|---|---|---|---|---|---|---|---|---|---|---|---|---|---|
> | Ours APT 1NFE | **82.00** | 84.21 | 73.15 | 61.89 | 26.11 | 71.78 | 23.20 | 92.41 | 62.78 | 63.04 | 90.20 | 41.19 | 69.05 | 96.38 | 97.26 | 98.57 | 98.54 | 88.98 | 19.31 |
> | Ours APT 2NFE | 81.85 | 84.39 | 71.70 | 59.60 | 25.70 | 71.95 | 23.88 | 91.30 | 68.61 | 62.68 | 87.00 | 40.10 | 69.60 | 95.89 | 96.54 | 98.48 | 98.49 | 85.92 | 18.84 |
> | Consistency 1NFE | 67.05 | 73.78 | 40.15 | 4.85 | 17.47 | 19.38 | 15.33 | 39.13 | 20.56 | 41.96 | 33.20 | 10.78 | 42.95 | 96.80 | 97.39 | 98.72 | 98.35 | 86.93 | 21.33 |
> | Consistency 2NFE | 74.20 | 78.83 | 55.69 | 23.78 | 22.88 | 44.21 | 20.96 | 63.34 | 33.33 | 54.12 | 63.60 | 24.48 | 59.93 | 96.21 | 96.33 | 98.55 | 98.29 | 88.88 | 19.20 |
> | Consistency 4NFE | 77.97 | 81.93 | 62.10 | 37.30 | 23.73 | 57.98 | 22.16 | 77.58 | 53.61 | 59.35 | 74.00 | 25.81 | 66.38 | 96.37 | 95.97 | 98.20 | 98.15 | 90.43 | 18.35 |
> |-|
> | Diffusion Base 50NFE | 82.15 | 84.36 | 73.31 | 71.68 | 26.09 | 74.95 | 23.87 | 90.49 | 75.56 | 62.94 | 89.60 | 33.63 | 69.85 | 97.02 | 96.44 | 97.61 | 97.54 | 88.69 | 18.93 |

---

> > ### Comment · Reviewer_afjC · 2025-04-04
> >
> > Thanks for the reply, the one-step results on the VBench look promising.

---

> > > ### Author Response · Authors · 2025-04-04
> > >
> > > Thanks for the encouraging assessment. Please let us know if there are any additional questions.
> > > We appreciate that the reviewer can update their review in light of this response.
> > >
> > > The Authors

---

### Official Review · Reviewer_Df84 · 2025-03-13

**Overall Recommendation:** 3

**Summary:**

The paper introduces Adversarial Post-Training (APT), a method that accelerates diffusion-based video generation from multiple inference steps to a single step while preserving high-quality visual output. The approach builds on a pre-trained diffusion model and uses direct adversarial training with real data.

**Claims And Evidence:**

The claim from Ln 42 to Ln 45, “It is important to notice the contrast to existing diffusion distillation methods, which use a pre-trained diffusion model as a distillation teacher to generate the target.” is not accurate. For example, UFOGen, SF-V do not require the pre-trained diffusion model to generate the target.

The claim from Ln 56 to Ln 57, “APT demonstrates the ability to surpass the teacher by a large margin in some evaluation criteria” seems conflicted with the experiments in Appendix Table 5.

**Essential References Not Discussed:**

Lai, Zhixin, Keqiang Sun, Fu-Yun Wang, Dhritiman Sagar, and Erli Ding. "InstantPortrait: One-Step Portrait Editing via Diffusion Multi-Objective Distillation." In *The Thirteenth International Conference on Learning Representations*.

Zhang, Zhixing, Yanyu Li, Yushu Wu, Anil Kag, Ivan Skorokhodov, Willi Menapace, Aliaksandr Siarohin et al. "Sf-v: Single forward video generation model." *Advances in Neural Information Processing Systems* 37 (2024): 103599-103618.

Mao, Xiaofeng, Zhengkai Jiang, Fu-Yun Wang, Wenbing Zhu, Jiangning Zhang, Hao Chen, Mingmin Chi, and Yabiao Wang. "Osv: One step is enough for high-quality image to video generation." *arXiv preprint arXiv:2409.11367* (2024).

Wang, Fu-Yun, Zhaoyang Huang, Alexander Bergman, Dazhong Shen, Peng Gao, Michael Lingelbach, Keqiang Sun et al. "Phased Consistency Models." *Advances in Neural Information Processing Systems* 37 (2024): 83951-84009.

**Experimental Designs Or Analyses:**

Comparisons with other distillation methods would be helpful, (e.g. ADD, LADD, UFOGen).

**Methods And Evaluation Criteria:**

While the paper claims superior video generation quality, most quantitative comparisons are conducted against image models in the image domain. A user study and evaluations on benchmarks like VBench comparing the method with other open-source text-to-video generation models (e.g. opensora, hunyuan video, and etc.) would better demonstrate its effectiveness.

**Other Comments Or Suggestions:**

None.

**Other Strengths And Weaknesses:**

None.

**Questions For Authors:**

None.

**Relation To Broader Scientific Literature:**

Adversarial post-training to reduce the number of inference steps of diffusion models has been widely studied.

**Theoretical Claims:**

Looks good to me.

---

> ### Author Rebuttal · Authors · 2025-04-01
>
> **Related Works**
>
> We appreciate the reviewer pointing out the related works. We will add all of them in the revised paper.
>
> * SF-V and OSV are 1-step image-to-video generation. We initially did not include them because our work focuses on text-to-video generation. Both of these works are based on UNet based Stable Video Diffusion and can generate up to 1024x576 for 14 total frames. In comparison, our works can generate 1280x720 of total 48 frames (2s 24fps) from text prompts.
>
> * PCM is an adversarial consistency distillation approach very similar to Hyper and our paper has conducted extensive comparison with Hyper. We notice that the paper has also experimented with distillation on AnimateDiff along with AnimateLCM and AnimateDiff-Lightning. We will add this paper to our related works.
>
> * InstantPortrait focuses on 1-step image editing tasks with contributions primarily addressing task-specific problems, which may not be directly relevant to our problem.
>
>
> **Comparison Against Other Distillation Methods**
>
> The reviewer suggested a comparison against ADD, LADD, and UFOGen. In fact, both ADD and LADD are already included in our submitted paper. Specifically, the ADD method is referred to as SDXL-Turbo and the LADD method as SD3-Turbo, following the naming conventions used by Stability AI [[1](https://stability.ai/news/stability-ai-sdxl-turbo),[2](https://arxiv.org/abs/2403.12015)]. We apologize for any confusion and will clarify this in the revised version. We did not include UFOGen in our comparisons, as it was shown to performce worse than DMD2 in the DMD2 paper. Since our paper includes DMD2 as a baseline, we believe this provides a sufficient point of comparison.
>
>
>
> **VBench Results**
>
> We have provided VBench metrics for the rebuttal. Please refer to the rebuttal response to `Reviewer afjC`. These results will be included in the revised version of the paper.
>
> **Other Clarifications on Our Claims**
>
> About Ln42-Ln45: We have acknowledged in Ln70-Ln76 that UFOGen is closest to our work which only applies adversarial training on real data. We also elaborated our differences compared to UFOGen. Specifically, our discriminator design is closer to the traditional GAN, and our APT model can surpass the teacher model in some criteria which validates the proposed adversarial post-training.
>
> About Ln56-Ln57: We have elaborated in Appendix B how the traditional COCO FID and PFID metrics in Table 5 and 6 do not fully capture the model performance and thus opt for human evaluation. We leave the exploration for better automated metrics to future works. Our claim is based on Table 1 and 4, where our APT model shows improved visual fidelity compared to the original diffusion model.

---

> > ### Comment · Reviewer_Df84 · 2025-04-01
> >
> > Thanks for the prompt reply by the authors. Most of my concerns were addressed. My apologies for the confusion caused by `Comparisons with other distillation methods would be helpful, (e.g. ADD, LADD, UFOGen)`. My initial thoughts were since this paper focuses on video generation, including such comparisons in video space on the same video model might help the readers have a better understanding of the effectiveness of the proposed method.

---

> > > ### Author Response · Authors · 2025-04-01
> > >
> > > We sincerely appreciate the review and the response.
> > >
> > > We would like to note that existing methods such as ADD and LADD have several limitations for video tasks. For example, LADD requires pre-generating videos from the teacher model, which is computationally expansive especially for the high-resolution video generation tasks. ADD uses DINOv2 discriminator on the pixel space which requires decoding and backpropagation through the VAE decoder. This is also computationally and memory-wise infeasible. These limitations prevented the use of these methods for video generation and inspired our design of APT which is more suitable for the video task.
> > >
> > > Please feel free to let's us know if there are any more questions.

---

### Official Review · Reviewer_RHEj · 2025-03-14

**Overall Recommendation:** 3

**Summary:**

The paper presents a post-training approach to transform a pretrained video diffusion model (based on DiT architecture) into a one-step generation model, unlike traditional diffusion models requiring multiple (or at least a few) steps. Unlike many existing distillation methods that train a separate student model under the supervision of a teacher model, this work directly applies post-training to a pretrained model using adversarial training (GAN-based framework). Notably, the authors introduce techniques to stabilize training, particularly an approximated version of R1 regularization, as the standard R1 loss is unsuitable for large-scale training scenarios.

**Claims And Evidence:**

The main claims include:

- Effective conversion from multi-step diffusion to one-step generation through adversarial post-training.
- Improved training stability due to the introduced approximated R1 loss.

The authors provide qualitative evidence supporting the overall effectiveness of their final model. However, there is a gap: the paper lacks direct comparison between the adversarially post-trained model and its initial consistency-distilled baseline. This omission makes it difficult to clearly attribute performance gains to the proposed adversarial post-training strategy alone.

**Essential References Not Discussed:**

No essential missing references identified.

**Experimental Designs Or Analyses:**

While the experimental design convincingly shows the practical engineering value and demonstrates stability gains from the approximated R1 loss, it lacks thorough comparative analyses. Specifically, the paper fails to provide direct visual and quantitative comparisons between the final adversarially post-trained model and the initial consistency-distilled baseline.

**Methods And Evaluation Criteria:**

The proposed adversarial post-training method and the approximated R1 regularization are well-chosen and relevant to the stated problem. Evaluation criteria (qualitative visual comparisons with state-of-the-art methods) align with standard practice. However, the evaluations suffer from a crucial shortcoming: absence of direct comparative evaluation against the initial consistency-distilled model, undermining the strength of claims regarding performance improvements.

**Other Comments Or Suggestions:**

To significantly strengthen the paper, the authors should provide explicit qualitative and quantitative comparisons between the final adversarially post-trained model and the initial consistency-distilled baseline, clarifying whether performance genuinely improves.

**Other Strengths And Weaknesses:**

**Strengths:**

- Practical approach achieving effective one-step video generation.
- Useful technical innovation in approximated R1 regularization, addressing large-scale training stability issues.

**Weaknesses:**

- Inadequate comparative analysis against the baseline consistency-distilled model.
- Limited novelty regarding model initialization approach relative to existing methods (e.g., DMD).

**Questions For Authors:**

1. Could you provide direct qualitative and quantitative comparisons between your final adversarially post-trained model and the initial consistency-distilled baseline? This is crucial for evaluating the genuine effectiveness of your proposed method.

2. Given that the adversarial post-training starts from an already strong consistency-distilled model, can you explicitly demonstrate if and how the proposed method improves video quality rather than merely preserving it?

3. Could the authors elaborate further on the fundamental difference between their post-training approach and existing distillation methods like DMD, which also initialize student models with teacher weights?

**Relation To Broader Scientific Literature:**

The paper relates closely to recent literature on accelerated diffusion models and distillation techniques. However, the distinction made by authors between their post-training approach and existing teacher-student approaches (e.g., DMD, CVPR 2024) is somewhat ambiguous since many student models are similarly initialized from teacher weights.

**Theoretical Claims:**

No explicit theoretical claims or proofs are provided.

---

> ### Author Rebuttal · Authors · 2025-04-01
>
> **Comparison with Consistency Baseline**
>
> We respectfully point out that consistency distillation (CD, including methods such as LCM) has been extensively studied in prior works (e.g., DMD, DMD2, Lightning, Hyper-SD, LADD), which consistently show that CD struggles to produce sharp results in a single step. For instance, see Figure 3 in [Lightning](https://arxiv.org/pdf/2402.13929). This has also been shown in Figure 9 of our submission which illustrates that as training progresses, the initial CD model generates noticeably blurry outputs in one step. Therefore, in our work, we choose to compare directly against state-of-the-art methods rather than the CD baseline.
>
> However, following the reviewer's suggestion, we have included the CD-initialized model as a baseline. We compute metrics: FID, PFID, and CLIP scores, on both the COCO-10K and COCO-5K benchmarks. As shown in the table below, the results are consistent with those previously reported in the paper. We will incorporate them in the revised paper.
>
> In addition, we have included a video comparison between the CD and APT models on the VBench benchmark. Please refer to our response to `Reviewer afjC` for the detailed results.
>
> | *COCO-10K*         | FID↓   | PFID↓  | CLIP↑ |
> |-------------------|-------:|-------:|------:|
> | Diffusion 25step  | 20.7   | 24.7   | 33.1  |
> | Consistency 1step | 114.1  | 161.3  | 22.3  |
> | APT 1step         | 22.1   | 28.5   | 32.2  |
>
> | *COCO-5K*          | FID↓   | PFID↓  | CLIP↑ |
> |-------------------|-------:|-------:|------:|
> | Diffusion 25step  | 26.9   | 30.6   | 33.2  |
> | Consistency 1step | 119.6  | 164.6  | 22.3  |
> | APT 1step         | 27.9   | 34.6   | 32.3  |
>
> More visualization comparison between CD and APT is provided in this [external link](https://rebuttal.blob.core.windows.net/assets/comparison.html).
>
> **Differences to Existing Methods**
>
> We would like to elaborate on the differences between our methods and existing methods (e.g. DMD CVPR24):
>
> 1. **Simplicity**: DMD combines variational score distillation (VSD) and rectified flow (RF) objectives, as VSD alone can lead to mode collapse while RF can cause sharpness issues. DMD2 changes to use both VSD and adversarial objectives, but this setup requires training 3 networks: the student, the VSD negative score model, and the adversarial discriminator. In contrast, our method adopts a simpler approach, using only an adversarial objective with CD initialization. This streamlined design allows the model to adapt quickly—in just 350 iterations (see Figure 9 in the paper).
>
> 2. **Post-training vs. distillation quality**: Existing methods (such as DMD, DMD2, Lightning, Hyper, ADD, and LADD) distill the teacher results, where the teacher model is the quality upper-bound. Our method is adversarial post-training (APT) against real data, and we demonstrate that it is able to surpass the teacher diffusion model in some criteria. The discussion is provided in Appendix H.
>
> 3. **First large-scale video generation with 1NFE**: Existing methods like DMD and LADD require precomputing teacher noise-sample pairs, which can be computationally expensive—especially for high-resolution video tasks. In contrast, our method trains directly on real data, avoiding these overheads. This enables us to be the first to demonstrate one-step generation using a large-scale T2V model, achieving 1280×720 resolution at 24 fps for 2-second videos.
>
> 4. **Closer to traditional GANs**: Existing adversarial distillation methods generally follow the DiffusionGAN approach, which may appear similar to GANs but differs in key ways: 1) noise corruption is added to the discriminator inputs (e.g., LADD, UFO-GAN, Lightning), and 2) it freezes the discriminator backbone and only trains the generator (e.g., ADD, LADD). In contrast, our method more closely resembles classical GANs: the discriminator is fully trainable, and no noise corruption is applied to its inputs. Additionally, we introduce an approximate R1 loss, inspired by traditional R1 regularization, which significantly improves training stability. We hypothesize that this design contributes to the reduction of visual artifacts in our one-step generation, as compared to existing works (LADD, Lightning, etc.), as illustrated in Figure 3.
>
> We hope the above points clarify the differences from related work and substantiate the contributions of our method.

---

### Decision · Program_Chairs · 2025-05-01

**Decision:**

Accept (poster)

**Comment:**

The paper received 2 Weak Accept and 1 Weak Reject score pre-rebuttal. The scores are not changed after the rebuttal. Although Reviewer afjC did not change his Weak Reject score, his response to the rebuttal was positive, and the rebuttal seemed to address his concern well.

The ACs checked and found the paper's results strong and valuable to the community. Hence, we agreed to accept the paper to ICML. The authors should include the discussions and results in the rebuttal, particularly the additional VBench results, in the camera-ready version.